# Simple, Accurate, and Efficient Axis-Aligned Decision Tree Learning

## Abstract

Decision Trees (DTs) are widely used in various domains for their simplicity and interpretability. However, traditional DTs often suffer from low accuracy and reduced robustness because they rely on fixed splits and a greedy approach to decision-making. While recent approaches combining decision trees with optimization seek to balance accuracy, computational efficiency, and interpretability, they still fall short in certain aspects. In this paper, we introduce a novel **Pro**babilistic **u**nivariate **D**ecision **T**ree (*ProuDT*), a non-greedy, axis-aligned tree that aims to address these challenges and achieve significant improvements. By assigning a single deterministic feature to each decision node, *ProuDT* ensures univariate splits while preserving the differentiability of soft decision trees for gradient-based optimization. This tree enhances interpretability through transparent feature utilization in decision-making. Additionally, *ProuDT* simplifies the optimization process and reduces computational cost by avoiding complex parameters. Extensive experiments on tabular datasets demonstrate *ProuDT*'s superior performance and scalability, particularly in binary and multi-class classification tasks.

## 1 Introduction

Decision trees are among the most widely used and easily understood methods in supervised learning. The classic decision trees include CART (Breiman et al., 1984) and C4.5 (Quinlan, 2014), which continues to be the most widely recognized trees and are the basis of various advancements. The simplicity and interpretability of decision trees make them a favored choice for many applications. Current approaches to learning decision trees are categorised into greedy and non-greedy methods. Through greedy optimization (Mingers, 1989; Raileanu & Stoffel, 2004), trees are grown following one specific criterion (e.g., Gini impurity) at decision nodes. The process happens recursively with further nodes being split without reconsidering previous splits during tree induction. The greedy nature usually leads to sub-optimal solutions (Cormen et al., 2022). To address sub-optimal problems, non-greedy methods have been extensively explored from various perspectives. Global tree search approaches, such as lookahead methods (Norton, 1989; Kiossou et al., 2024), evolutionary algorithms (Barros et al., 2011), or mathematical programming techniques like mixed-integer programming (Günlük et al., 2021; Bertsimas & Dunn, 2017; Bertsimas et al., 2022), are all optimization techniques. However, a key limitation of these global search approaches is their lack of scalability. Exploring the entire tree space dramatically increases computational cost, making such methods feasible only for small trees and datasets.

Due to the scalability limitations of global search methods and the aim for high classification accuracy, alternative optimization techniques have been explored. Among these alternatives, gradient-based optimization has gained increasing popularity. Compared to greedy trees with hard split, these trees demonstrate better learning capability especially with probabilistic splits, i.e., soft splits, and have good prediction performance (Frosst & Hinton, 2017; Wan et al., 2020). However, there are interpretability, accuracy, and computational cost limitations among existing efforts. Probabilistic trees often use multivariate splits, which reduces interpretability by making the roles of features at decision nodes unclear. To address the reduced interpretability caused by multivariate splits, researchers have attempted to maintain univariate splits in soft trees using gradient-based optimizations for hard splits (Marton et al., 2024). However, these efforts negatively impact classification accuracy and limit scalability. Moreover, the computational and memory costs of dense feature rep-

resentations are high due to the many learnable parameters involved in gradient-based optimization. For users seeking a simple and transparent model for classification tasks, the complexity can be a significant drawback. This may explain why traditional decision trees remain popular, as their simplicity is appealing despite their potential limitations.

Existing approaches to optimizing decision trees for high accuracy often increase model complexity, which in turn affects interpretability. Specifically, most gradient-based methods struggle to ensure sparse features at decision nodes, making it difficult to maintain transparency in the decision-making process. Three commonly explored methods aim to address this challenge. The first approach involves growing an oblique tree to improve accuracy and then applying post-hoc feature removal strategies during prediction. However, this creates a discrepancy between the features used during training and testing, which ultimately degrades the model's performance. The second approach seeks to learn the position of individual features during tree growth. This is done using a dense matrix to store all features across all decision nodes, aiming to identify the most suitable splitting feature through parameter learning. While this method improves flexibility, it significantly increases memory and computational costs, limiting scalability for higher dimensions and deeper trees. The third approach employs traditional greedy search algorithms, such as computing information gain. While this method easily identifies a deterministic feature for splitting at each decision node, its performance is often compromised by the inherent limitations of the greedy strategy.

To address these challenges in gradient-based tree construction, we propose the Probabilistic univariate Decision Tree (*ProuDT*) to jointly improve: (1) accuracy, (2) convergence and inference speed by reducing computational cost through fewer parameters, and (3) interpretability by offering insights into feature utilization and decision-making process. Specifically, our contributions are:

1. We introduce an effective and efficient probabilistic tree learning strategy with fewer learnable parameters involved during the tree induction. Rather than explicitly focusing on learning the single feature for node splitting, our approach naturally achieves transparent and effective single-feature splits. To the best of our knowledge, this is the first method to directly utilize univariate splitting for probabilistic tree induction.

2. We propose a simple yet powerful axis-aligned decision tree that achieves high accuracy and scalability, outperforming leading univariate trees. Moreover, by using the default settings for a robust tree learning, users can bypass the complexity of hyperparameter tuning, making the tree easy to deploy.

3. We offer insights into feature's utilization that contribute to the decision-making process.

## 2 RELATED WORKS

**Greedy trees:** Several well-established decision trees have been developed for decades, e.g., QUEST (Loh & Shih, 1997), CART (Breiman et al., 1984), C4.5 (Quinlan, 2014), CHAID (Kass, 1980). The most prominent among them is the CART and C4.5, which construct trees by recursively splitting the data based on feature values that minimize impurity or maximize information gain. Due to the greedy hard splitting nature, these trees suffer from sub-optimal problems that affect accuracy.

**Non-greedy trees:** To handle the sub-optimal issue, non-greedy works have been proposed from different aspects. MurTree (Demirović et al., 2022) applies dynamic programming and search algorithms to generate optimal decision trees to handle categorical features. Nunes et al. (2020) proposes the Monte Carlo Tree Search (MCTS) algorithm to facilitate lookahead tree navigation and overcome the sub-optimal problem. Mixed-integer programming is also commonly explored (Bertsimas et al., 2022; Zantedeschi et al., 2021). However, these methods introduce scalibility and computation concerns. Tree alternating optimization (TAO) iteratively optimize node parameters and enable the construction of sparse oblique decision trees (Carreira-Perpinán & Tavallali, 2018). Through gradient-based optimization, DGT (Karthikeyan et al., 2021) achieves an oblique tree with hard splits. In contrast, we aim to optimize a univariate tree, as it can be more interpretable than multivariate trees.

**Probabilistic Trees:** Probabilistic trees soften the path routing with probabilistic splits, where an instance can be routed along multiple branches with certain probabilities, typically using functions like sigmoid to model the smooth transition (Irsoy et al., 2012; Norouzi et al., 2015; Lee & Jaakkola, 2019). The prediction is determined either by an aggregation of all leaves, for example through a

weighted sum (Hehn et al., 2020; Suarez & Lutsko, 1999; Irsoy et al., 2012), or by the leaf with the highest path probability (Frosst & Hinton, 2017). With the growing interest in explainability in neural networks, the combination of neural networks and decision trees has become increasingly popular (Frosst & Hinton, 2017; Yang et al., 2018; Wan et al., 2020; Rodríguez et al., 2024). However, these probabilistic trees often incorporate a linear combination of input features when determining their splits. These multivariate trees preclude users from understanding the feature utilization. Although researchers try to make the tree as interpretable as possible (Xu et al., 2022; Silva et al., 2020), it is still a challenge to clarify the individual feature's contribution. (Silva et al., 2020) trained a probabilistic tree with multivariate splits and then, during prediction, transformed the tree into a univariate tree by selecting the single feature with the highest weight from the multiple features at each node. However, this process led to an inconsistency between the feature splits used in training and testing, causing accuracy loss during prediction.

**Univariate decision trees:** The majority of gradient-based approaches focus on trees with multivariate or more complex splits. This stands in contrast to learning trees with univariate tests. Therefore, constructing a univariate tree at the training stage is a potential solution to overcome the weakness of multivariate trees. DNDT (Yang et al., 2018) leverages soft binning to make splits and yields soft and axis-aligned tree. However, as the author mentioned, DNDT cannot scale well to datasets with large number of features, i.e., more than 12, due to the limitation of Kronecker product. Zantedeschi et al. (2021) utilizes the Argmin Differentiation to optimize tree parameters. This method struggles with scalability to larger feature spaces. Recently, Marton et al. (2024) propose GradTree, a method that enables hard splitting via gradient optimization. In backward propagation, to overcome the non-differentiable process, by leveraging straight-through operator, they bypass the non-differentiable function of hard split derived from forward propagation. However, mismatched forward and backward propagations introduce gaps between training and testing stage, leading to potential accuracy loss. In univariate tree induction, a key consideration is selecting the single feature for splitting at each node. GradTree uses all input features at each node, following the probabilistic tree's method, but introduces the dense matrix to store all feature's weights (indices) and thresholds across all nodes for single-feature selection. This makes the method computationally expensive and memory-intensive, particularly for high-dimensional datasets or deeper trees. The trade-off between optimization flexibility and scalability limits the model's applicability to large-scale tasks. In contrast, our proposed univariate tree induction strategy simplifies the optimization process while maintaining efficiency and interpretability and can yield superior prediction result than the leading univariate trees.

## 3 METHODOLOGY

We introduce *ProuDT* in this section. Unlike classical decision trees, *ProuDT* employs a probabilistic routing strategy to yield more robust and accurate result. Univariate splitting and feature selection is introduced in section 3.1. We describe the tree learning in section 3.2.

### 3.1 UNIVARIATE SPLITTING

In the probablistic decision tree induction, the sigmoid function is applied to turn hard splits into weighted ones (soft splits). The output $\sigma(f(\boldsymbol{x})$(or $(1 - \sigma(f(\boldsymbol{x}))$ represents the probability $p$ of a sample $\boldsymbol{x}$ going left (or right). A commonly adopted strategy involves using a linear combination of input features for splitting during training, followed by seeking sparse features for prediction. At each decision node, the probability is then computed using a sigmoid function, with weights applied to all input features. The probabilistic trees mentioned above (section 2) treat both weights and thresholds as learnable parameters per node. As show in Eq.1, for a given input feature vector $\boldsymbol{x}$, the bias $b_i$ and corresponding weight vector $\boldsymbol{w}_i$ are learned at the splitting node $i$. GradTree (Marton et al., 2024) improves this strategy by introducing a dense matrix to store each threshold for each feature at each node, instead of using a general bias, with the goal of aligning specific features with their thresholds. However, this method increases memory and computational costs, while also introducing non-differentiable functions.

$$p_i(\boldsymbol{x}) := \frac{1}{1 + e^{-(\boldsymbol{x} \cdot \boldsymbol{w}_i) - b_i))}} \tag{1}$$

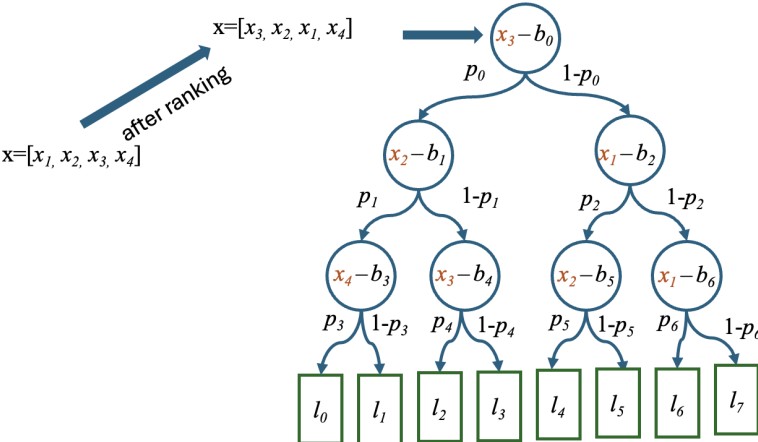

Figure 1: *ProuDT* construction. We show the feature assignment in *ProuDT* architechture for an input sample vector $x$ consisting of four features. After ranking features from an original order $x_1, x_2, x_3, x_4$ to a ranked order $x_3, x_2, x_1, x_4$, we position the features to decision nodes in a cyclic fashion. The branches' probabilities from decision node $i$ are determined by the individual feature and the only learnable bias $b_i$ of that node.

Unlike the approach proposed in GradTree, we take a different path by directly assigning a deterministic feature to each splitting node. This approach allows us to explore the learning potential of individual features at each node (Eq.2). We use mutual information to rank features before constructing the tree. During training, individual features, ranked from most to least important, are assigned to each node in a cyclic order, starting from the root node and continuing through the internal nodes up to the penultimate depth (see Figure 1).

Additionally, we design the splitting criteria by treating only the threshold as a learnable parameter. Our goal is not only to reduce memory usage by using fewer parameters, but also to maintain the interpretability of traditional decision trees. Consequently, our probabilistic tree, *ProuDT*, retains deterministic feature splits similar to those in traditional decision trees, while offering the potential for improved learning and predictive performance due to its "soft" properties.

$$p_i(\boldsymbol{x}) := \frac{1}{1 + e^{-(x_{ji} - b_i)}} \tag{2}$$

Where,

- $j$ denotes the index of the employed feature.
- $x_{ji}$ denotes the individual feature utilized for splitting at node $i$.

## 3.2 TREE LEARNING

To simplify the process, *ProuDT* focuses on fully constructed binary trees for classification problems. During the initialization stage, the threshold value at each decision node is treated as a learnable parameter and is initialized at the beginning. Additionally, each leaf node contains a vector of class distribution scores, which are also initialized as learnable parameters. Individual features are allocated to each decision node as described in section 3.1.

During the training stage, in the forward splitting, each decision node computes a probability for the left branch using the output of a sigmoid function, while the right branch receives the complement of this probability. For the entire tree, each path's probability is the product of the branch probabilities along that path, denoted as $P_l$. The path probability is then multiplied by class distribution score $\lambda_l$ at the corresponding leaf $l$ to obtain the final value. Finally, with the aggregation of all weighted leaf nodes (Eq.3), the prediction is made. $\phi$ denotes the weighted sum of class scores across each leaf node $l$, and $Q_k$ is the predicted probability for class $k$ after applying the softmax function (Eq.4).

$$\phi = \sum_{l=0}^{2^d-1} \lambda_l P_l(\boldsymbol{x}) \tag{3}$$

$$Q_k = \frac{\exp(\phi_k)}{\sum_{k'} \exp(\phi_{k'})} \tag{4}$$

In the backward propagation, the learning parameters are updated using their gradients computed from the loss function (Eq. 5).

For a single training sample with input vector $\boldsymbol{x}$, the focal loss function is:

$$\mathcal{L}_{\text{focal}}(\boldsymbol{x}) = -\sum_{k=1}^{C} T_k (1 - Q_k)^\gamma \log(Q_k) \tag{5}$$

Where:

- $\mathcal{L}_{\text{focal}}(\boldsymbol{x})$ is the focal loss for input $\boldsymbol{x}$,
- $C$ is the number of classes,
- $T_k$ is the true label indicator for class $k$ (1 if class $k$ is the true class, 0 otherwise).
- $\gamma \geq 0$ is the focusing parameter,

### 3.3 INTERPRETABILITY OF FEATURE UTILIZATION

The interpretability of our tree stems from the clear ranking and structured positioning of features during the learning process. By highlighting the factors that contribute most to decision-making, our tree provides valuable insights into the key features driving predictions. Its strong predictive performance reinforces the confidence in feature importance derived from mutual information. This structured approach provides a transparent explanation of feature significance and decision pathways, offering clarity and assurance in understanding which features play the most critical roles in the decision-making process.

## 4 EXPERIMENTS

In this section, we compare our tree with open-source GradTree[1], the state-of-the-art non-greedy univariate tree, and CART, the standard univariate tree. We conduct experiments on both numerical and categorical datasets, comparing performance across datasets with varying sample sizes, feature dimensions, and class distributions. We find our *ProuDT* outperforms the other leading models. Besides, *ProuDT* is faster than GradTree in terms of the tree induction and inference time. Additionally, we conduct ablation studies to check the feature selection strategy and verify the loss design. We provide the source code[2] for reproducibility.

### 4.1 EXPERIMENTAL SETUP

**Preliminary Study:** For any gradient-based tree, the tree depth should be specified in advance (Frosst & Hinton, 2017; Blanquero et al., 2021; Verwer & Zhang, 2019), as the tree model requires initialization in the beginning. Hence, the depth of a tree is a crucial hyperparameter. Fixing the tree depth, rather than performing an automatic depth search, is more practical for tree training. Since optimizing a fixed structure is already computationally expensive, iterating over multiple structures is not prioritized (Costa & Pedreira, 2023).

To gain insights into optimal default depth setting for any given datasets in *ProuDT*, we conducted a preliminary study on 12 UCI datasets of varying sample size, dimensionalities and class numbers.

---

[1]https://github.com/s-marton/GradTree
[2]https://github.com/Alicesn/ProuDT

We evaluated the prediction performance across tree depth ranging from 2 to 14 (see details in Table 4 and results in Figure 3 in Appendix). From the result of the preliminary study, we found that our tree (1) does not overfit even as the tree grows deeper and (2) achieves optimal performance at shallow depths (below 9) for most datasets, while deeper depths (above 9) are better suited for large, high-dimensional datasets. Most datasets in the preliminary study reach optimal accuracy between depths 2 and 5. Beyond this point, the accuracy curve remains flat as the depth increases, with no signs of overfitting.

**Datasets and Preprocessing in Experiment:** The formal experiments are conducted on another set of 12 multiple UCI datasets, specifically, two small multi-class datasets from Yang et al. (2018), four large high-dimensional multi-class datasets from Karthikeyan et al. (2021) and Marton et al. (2024), and six large binary datasets from Marton et al. (2024). Both categorical and numerical datasets are included. All datasets in these experiments are classification problems. For all datasets, we follow the data preprocessing from Popov et al. (2019) and apply the quantile transform[3] to convert each data feature into a normal distribution. We applied a 80%/20% train-test split. Additionally, 20% of the training data was set aside as validation data to control early stopping at optimal epoch.

**Hyperparameters and Default Setting in Experiment:** We view our model as a user-friendly method suitable for non-expert users. Specifically, we refrain from adjusting hyperparameters, opting to use the default settings. Our focus is on implementation simplicity, providing a high-performance method that is easy to use.

Based on the insights gained from the preliminary study, we selected a depth of 8 as the default for low-dimensional datasets and a depth of 11 for high-dimensional datasets (feature size $> 100$) in our formal experiment. Although these default depths are slightly deeper than observed in the preliminary study, our goal is to achieve more robust performance across diverse, unknown datasets. We test our tree using the 12 standard datasets, as described before. For GradTree, we implement its suggested default depth 10. For CART, we employ the standard sklearn implementation. To ensure a fair comparison with the most recent state-of-the-art univariate GradTree, we conducted all experiments using a NVIDIA A100-PCIE-40GB GPU. It is worth noting that *ProuDT* is not only efficient but also easy to deploy on both CPU and GPU systems, as it does not involve complex computations. This makes our approach computationally lightweight and scalable across different hardware setups.

## 4.2 RESULT

We compare accuracy (also see Table 5 in Appendix for additional F1-score with similar comparisons), as well as training and testing time for evaluating different trees. In terms of accuracy (Table 1), our method outperforms the most recent state-of-the-art non-greedy tree model and the standard CART on binary datasets (i.e., $N_c = 2$). *ProuDT* also achieves competitive accuracy compared to CART and outperforms GradTree on multi-class datasets, particularly in high-dimensional scenarios.

Regarding training time (see details in Table 6 in Appendix), the greedy CART undoubtedly remains the fastest, with training times typically under 1 second. Optimization-based solutions tend to require longer training times. For most binary datasets, *ProuDT* completes training in less than 20 seconds, whereas GradTree takes significantly longer, with its training time being heavily dependent on the sample size. For example, on the BANK MARKET dataset, which has the largest sample size, GradTree requires 240 seconds, while *ProuDT* completes in just 13 seconds. As the number of classes increases, *ProuDT* takes more time to converge. Notably, for the SEMEION dataset, which has 256 features, our tree utilized the default depth of 11. Although this deeper depth resulted in a longer training time (596 seconds), it achieved an accuracy improvement of 15% over CART and 30% over GradTree. In terms of test time, *ProuDT* averages 0.02 seconds across all datasets, compared to 0.34 seconds for GradTree. Non-greedy CART remains the fastest, with an average test time of just 0.0005 seconds.

It is noteworthy that we set the default depth 8 in the experiment although less than depth 5 is suggested from the insights of the preliminary study. *ProuDT* would be faster with a reduced depth and can reach similarly competitive accuracy.

---

[3]sklearn.preprocessing.QuantileTransformer

Table 1: Accuracy comparison of *ProuDT*, *GradTree*, and *CART* on various classification datasets. We provide the datasets' structure, i.e., sample size ($N_s$), feature size ($N_f$), and class size ($N_c$). The results are shown as average±std over 10 runs. The bold font indicates the best result.

| Dataset | $N_s, N_f, N_c$ | ProuDT (%) | GradTree (%) | CART (%) |
|---|---|---|---|---|
| ADULT | 32561, 14, 2 | **84.50±0.45** | 81.31±2.03 | 80.90±0.47 |
| BANK MARKET | 45211, 14, 2 | **88.76±0.14** | 86.29±1.38 | 82.30±0.42 |
| CREDIT CARD | 30000, 23, 2 | **81.57±0.59** | 76.04±2.05 | 72.52±0.50 |
| RICE | 3810, 7, 2 | **92.35±0.66** | 92.13±1.10 | 88.48±0.79 |
| SPAMBASE | 4601, 57, 2 | **93.17±0.92** | 88.37±1.83 | 90.83±1.10 |
| MUSHROOM | 8124, 22, 2 | 99.74±0.18 | 97.33±7.16 | **99.90±0.14** |
| IRIS | 150, 4, 3 | **95.33±3.44** | 91.33±5.71 | 94.67±2.67 |
| SPLICE | 3190, 60, 3 | **90.13±1.34** | 83.32±5.23 | 89.14±1.79 |
| SEGMENT | 2310, 19, 7 | 95.45±0.88 | 88.64±3.58 | **96.45±0.53** |
| LETTER | 20000, 16, 26 | 87.62±0.68 | 48.92±3.09 | **88.03±0.29** |
| PENDIGITS | 10992, 16, 10 | **97.63±0.35** | 86.34±1.57 | 96.10±0.46 |
| SEMEION | 1593, 256, 10 | **87.11±1.58** | 48.93±3.60 | 74.59±2.00 |

## 4.3 ABLATION STUDY

### 4.3.1 FEATURE POSITIONING

In our tree design, we assign individual features to each decision node in a cyclic order after pre-ranking them. To evaluate the effectiveness of this feature positioning strategy, we compare the performance of mutual-information ranked feature order against the original feature order from the given datasets.

We conducted this experiment to evaluate test accuracy across different depths using the 12 UCI datasets from the preliminary study, analyzing the impact of depth variation on performance. Figure 2 shows that after rearranging ranked features to the nodes, with the most important feature being at the root node and the subsequently important features being positioned at deeper levels in sequence, performance across some datasets improves significantly at the very shallow depth such as depth 2 when compared to using a non-ranked initial feature sequence. However, as the depth increases, the performance gap between ranked and non-ranked feature positioning strategies narrows, demonstrating that our tree learning in a cyclic order is robust in achieving high accuracy regardless of different feature sequence at the decision nodes.

From this result, we gain insights that assigning features to decision nodes after feature ranking can contribute to tree's convergence. It is plausible as the importance at different decision node is different. The closer a decision node is to the top, the more decision paths it covers, and the greater its impact on the overall decision-making. Thus, we finalize our learning strategy by positioning pre-ranked features, obtained from any simple feature ranking technique, to contribute to the tree's performance at shallow depths.

### 4.3.2 LOSS DESIGN

We evaluate the effectiveness of focal loss in our tree model by comparing it with cross-entropy loss on those 12 datasets from the formal experiment, using the default depth setting. The results demonstrate the advantages of focal loss in terms of faster convergence and predictive capability. As shown in Table 2, focal loss accelerates convergence across all datasets when compared to cross-entropy loss. Meanwhile, as seen in Table 3, the accuracy for both loss functions remains comparable. The average accuracy achieved with focal loss is 90.93%, which is on par with the accuracy (91.08%) from cross-entropy loss.

## 4.4 DISCUSSION

*ProuDT* is user-friendly and ideal for non-experts to deploy with ease. Here are several takeways of our tree.

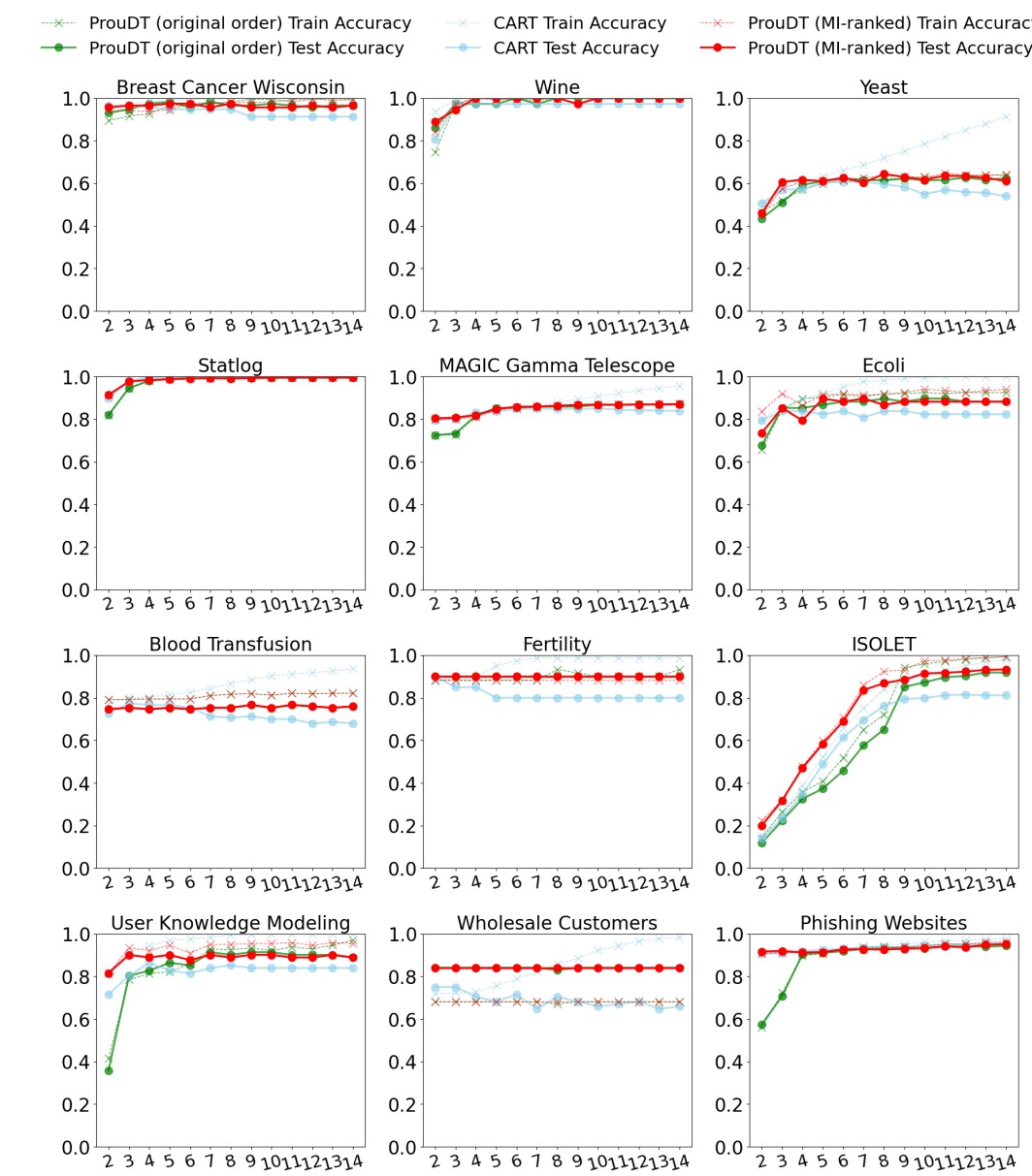

Figure 2: **Feature ranking power comparison.** The x-axis represents tree depth, and the y-axis represents classification accuracy (the higher, the better). Standard CART is included as a baseline for comparison. The performance of *ProuDT* is evaluated using both mutual-information ranked feature order and non-ranked original feature order across different datasets.

- Based on our extensive observations from the preliminary study (Figure 3 in Appendix), no overfitting is observed. Accuracy tends to plateau after reaching an optimal point at a lower depth, making it both simple and robust to employ a default depth setting.

- The results from the formal experiment demonstrate that *ProuDT* delivers superior accuracy performance and proves to be scalable across datasets of varying sizes.

- Our learning strategy is robust, enabling the use of different feature sequences at decision nodes. On one hand, feature ranking with simple techniques, such as mutual information or random forest, can accelerate convergence and achieves strong performance at lower tree depths. On the other hand, feature sequence is not necessary if users are not concerned with

Table 2: Training time comparison between focal loss (FL) and cross-entropy loss (CE) for *ProuDT* on various classification datasets. The average training time is reported over 10 trials.

| Dataset | $N_s$ | $N_f$ | $N_c$ | FL (s) | CE (s) |
|---|---|---|---|---|---|
| ADULT | 32561 | 14 | 2 | 15.68 | 41.44 |
| BANK MARKET | 45211 | 14 | 2 | 12.52 | 38.30 |
| CREDIT CARD | 30000 | 23 | 2 | 12.57 | 28.30 |
| RICE | 3810 | 7 | 2 | 11.23 | 23.13 |
| SPAMBASE | 4601 | 57 | 2 | 24.80 | 50.74 |
| MUSHROOM | 8124 | 22 | 2 | 22.16 | 53.71 |
| IRIS | 150 | 4 | 3 | 23.96 | 43.62 |
| SPLICE | 3190 | 60 | 3 | 39.89 | 70.12 |
| SEGMENT | 2310 | 19 | 7 | 91.56 | 165.04 |
| LETTER | 20000 | 16 | 26 | 399.09 | 441.38 |
| PENDIGITS | 10992 | 16 | 10 | 118.42 | 232.14 |
| SEMEION | 1593 | 256 | 10 | 596.45 | 919.18 |

Table 3: Accuracy comparison of focal loss (FL) and cross-entropy loss (CE) on various classification datasets.

| Dataset | $N_s$ | $N_f$ | $N_c$ | FL (%) | CE (%) |
|---|---|---|---|---|---|
| ADULT | 32561 | 14 | 2 | $84.50 \pm 0.45$ | $84.81 \pm 0.42$ |
| BANK MARKET | 45211 | 14 | 2 | $88.76 \pm 0.14$ | $88.99 \pm 0.17$ |
| CREDIT CARD | 30000 | 23 | 2 | $81.57 \pm 0.59$ | $81.82 \pm 0.44$ |
| RICE | 3810 | 7 | 2 | $92.35 \pm 0.66$ | $92.43 \pm 0.66$ |
| SPAMBASE | 4601 | 57 | 2 | $93.17 \pm 0.92$ | $92.92 \pm 0.65$ |
| MUSHROOM | 8124 | 22 | 2 | $99.74 \pm 0.18$ | $99.76 \pm 0.19$ |
| IRIS | 150 | 4 | 3 | $95.33 \pm 3.44$ | $96.00 \pm 2.91$ |
| SPLICE | 3190 | 60 | 3 | $90.13 \pm 1.34$ | $89.75 \pm 1.57$ |
| SEGMENT | 2310 | 19 | 7 | $95.45 \pm 0.88$ | $95.65 \pm 0.77$ |
| LETTER | 20000 | 16 | 26 | $87.62 \pm 0.68$ | $87.53 \pm 0.46$ |
| PENDIGITS | 10992 | 16 | 10 | $97.63 \pm 0.35$ | $98.01 \pm 0.37$ |
| SEMEION | 1593 | 256 | 10 | $87.11 \pm 1.58$ | $87.08 \pm 1.52$ |
| **Average** | | | | 90.93 | 91.08 |

the tree induction time. As tree depth increases, feature ranking becomes progressively less critical and eventually unnecessary.

## 5 CONCLUSION

We propose *ProuDT*, a simple yet powerful learning strategy to construct superior axis-aligned decision trees. In this work, we demonstrate that directly assigning individual features to the decision nodes in a cyclic order enhances both accuracy and efficiency in tree induction and inference. This method provides valuable insights into how individual feature utilization contributes to both interpretability and improved model accuracy. Moreover, experimental results from 12 datasets, complemented by a preliminary study on another set of 12 datasets, confirm the robustness and effectiveness of our default settings, further highlighting the ease of deployment and practical utility of *ProuDT*. Our exploration paves the way for developing more powerful axis-aligned trees following KISS principle.

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

## A APPENDIX

Table 4 shows the details of datasets for the preliminary study.

Table 4: Details of the 12 UCI datasets used in preliminary study

| Dataset | $N_s$ | $N_f$ | $N_c$ |
|---|---|---|---|
| Breast Cancer Wisconsin | 569 | 30 | 2 |
| Wine | 178 | 13 | 3 |
| Yeast | 1484 | 8 | 10 |
| Statlog | 58000 | 7 | 2 |
| MAGIC Gamma Telescope | 19020 | 10 | 2 |
| Ecoli | 336 | 8 | 8 |
| Blood Transfusion | 748 | 4 | 2 |
| Fertility | 100 | 9 | 2 |
| ISOLET | 7797 | 617 | 26 |
| User Knowledge Modeling | 403 | 5 | 5 |
| Wholesale Customers | 440 | 7 | 3 |
| Phishing Websites | 11055 | 30 | 2 |

We show the comparison of F1-score besides accuracy in Table 5.

Table 5: Comparison of F1-score for different trees across various datasets.

| Dataset | $N_s, N_f, N_c$ | ProuDT | GradTree | CART |
|---|---|---|---|---|
| ADULT | 32,561, 14, 2 | **0.774 ± 0.007** | 0.742 ± 0.046 | 0.740 ± 0.007 |
| BANK MARKET | 45,211, 14, 2 | 0.555 ± 0.024 | **0.629 ± 0.020** | 0.596 ± 0.008 |
| CREDIT CARD | 30,000, 23, 2 | 0.662 ± 0.015 | **0.671 ± 0.016** | 0.612 ± 0.007 |
| SPAMBASE | 4,601, 57, 2 | **0.928 ± 0.010** | 0.878 ± 0.019 | 0.904 ± 0.012 |
| RICE | 3,810, 7, 2 | **0.922 ± 0.007** | 0.920 ± 0.011 | 0.882 ± 0.008 |
| MUSHROOM | 8,124, 22, 2 | 0.997 ± 0.002 | 0.972 ± 0.077 | **0.999 ± 0.014** |
| IRIS | 150, 4, 3 | **0.953 ± 0.034** | 0.913 ± 0.057 | 0.946 ± 0.027 |
| SPLICE | 3,190, 60, 3 | **0.891 ± 0.015** | 0.809 ± 0.063 | 0.878 ± 0.021 |
| SEGMENT | 2,310, 19, 7 | 0.955 ± 0.009 | 0.884 ± 0.039 | **0.965 ± 0.005** |
| LETTER | 20,000, 16, 26 | 0.876 ± 0.007 | 0.479 ± 0.034 | **0.880 ± 0.003** |
| PENDIGITS | 10,992, 16, 10 | **0.976 ± 0.003** | 0.863 ± 0.016 | 0.961 ± 0.005 |
| SEMEION | 1,593, 256, 10 | **0.871 ± 0.016** | 0.482 ± 0.043 | 0.744 ± 0.019 |

Figure 3 presents the observations from a preliminary experiment conducted on 12 training datasets. The goal of this experiment was to gain insights into the default depth selection for our tree. The details of these 12 UCI datasets are included in the plot. We evaluated the model's prediction accuracy across various depths and found that it consistently achieves optimal accuracy at shallow depths for most datasets, with no signs of overfitting. For the extremely high-dimensional ISOLET dataset, we observed that a greater depth yields higher accuracy. Based on these findings, we recommend using a depth below 9 (such as depth 8) for most datasets. However, for high-dimensional datasets, a depth of 11 is advised.

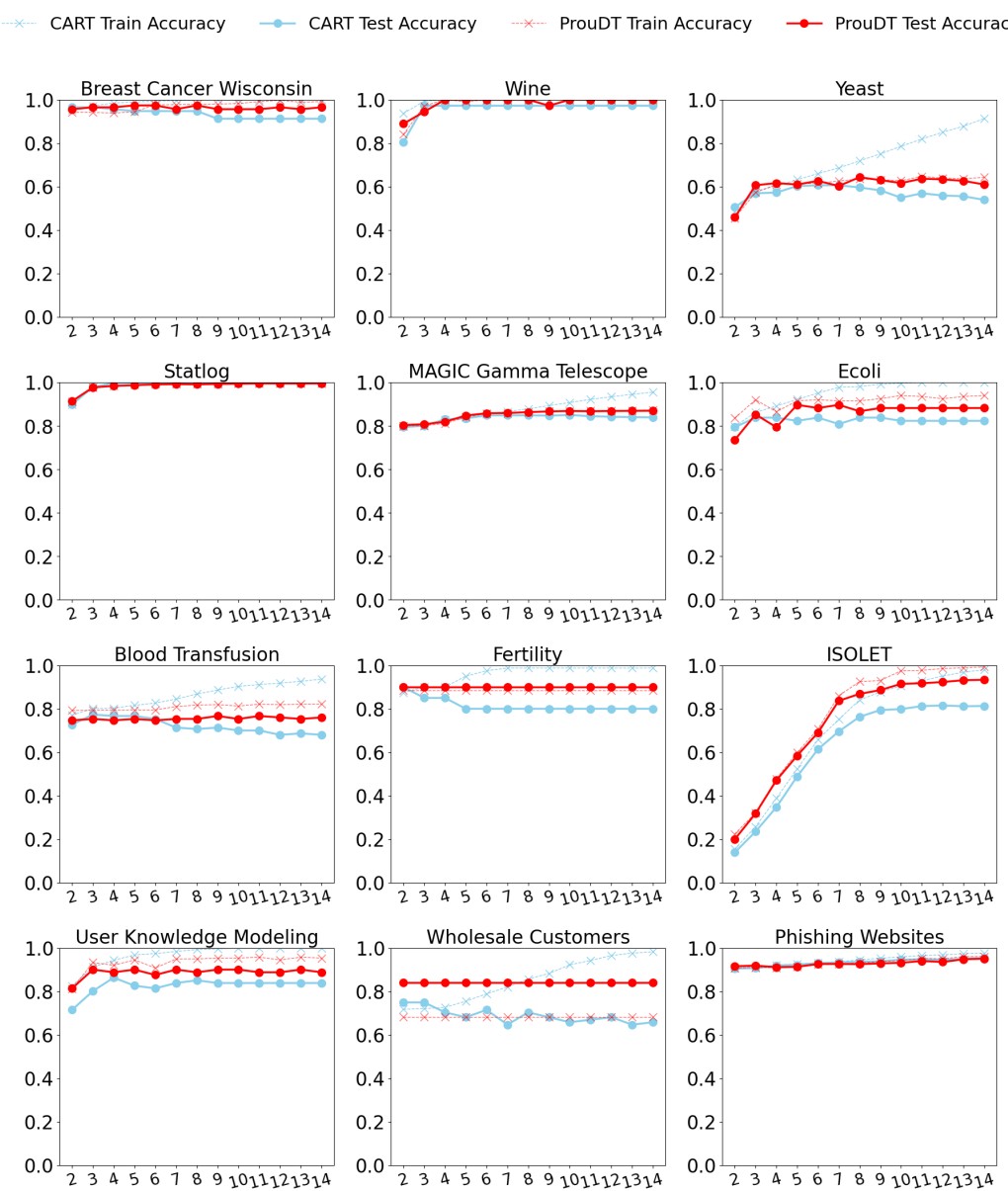

Figure 3: Preliminary study with *ProuDT* to assess the impact of tree depth on performance, with standard CART as a baseline. The x-axis represents tree depth, and the y-axis represents classification accuracy (the higher, the better). Both training and test accuracy are reported. *ProuDT* achieves optimal accuracy at shallow depths across most datasets, with the exception of the high-dimensional dataset (617 features), which requires a deeper depth. *ProuDT* outperforms CART at most depths.

Table 6: Training time (seconds) comparison of *ProuDT*, *GradTree*, and *CART* on experimental classification datasets. We provide the datasets' structure, i.e., sample size ($N_s$), feature size ($N_f$), and class size ($N_c$). The results are measured in seconds (s) across the models.

| Dataset | $N_s$, $N_f$, $N_c$ | ProuDT (s) | GradTree (s) | CART (s) |
|---|---|---|---|---|
| ADULT | 32,561, 14, 2 | 15.68 | 160.51 | 0.06 |
| BANK MARKET | 45,211, 14, 2 | 12.52 | 239.83 | 0.04 |
| CREDIT CARD | 30,000, 23, 2 | 12.57 | 144.73 | 0.55 |
| RICE | 3,810, 7, 2 | 11.23 | 40.99 | 0.02 |
| SPAMBASE | 4601, 57, 2 | 24.80 | 44.23 | 0.06 |
| MUSHROOM | 8,124, 22, 2 | 22.16 | 68.54 | 0.01 |
| IRIS | 150, 4, 3 | 23.96 | 29.74 | <0.01 |
| SPLICE | 3,190, 60, 3 | 39.89 | 30.97 | 0.02 |
| SEGMENT | 2,310, 19, 7 | 91.56 | 43.63 | <0.01 |
| LETTER | 20,000, 16, 26 | 399.09 | 124.97 | 0.06 |
| PENDIGITS | 10,992, 16, 10 | 118.42 | 89.29 | 0.05 |
| SEMEION | 1,593, 256, 10 | 596.45 | 35.83 | 0.02 |

