# OpenReview forum: "Simple, Accurate, and Efficient Axis-Aligned Decision Tree Learning"
_ICLR.cc/2025/Conference — ICLR 2025 Conference Withdrawn Submission_

### Official Review · Reviewer_rCK1 · 2024-10-23

**Soundness:** 2
**Presentation:** 3
**Contribution:** 2
**Rating:** 3
**Confidence:** 4

**Summary:**

This paper proposes a simple method to address challenges associated with feature selection for each splitting node and tree depth settings when learning axis-aligned decision trees using gradient-based methods. The paper demonstrates the effectiveness of the approach through experiments.

**Strengths:**

Training axis-aligned decision trees using gradient-based methods looks simple but difficult and important, as it offers potential benefits in terms of accuracy, computational cost, and interpretability. I believe that this simple yet effective approach could provide value to the research community. However, I feel that the evaluation supporting the effectiveness of the proposed method is currently insufficient. Strengthening this aspect would enhance the quality of the paper.

**Weaknesses:**

In Section 4.1, hyperparameters are set using the dataset employed for evaluation. This raises concerns about whether the settings are robust. If the intent is to highlight the general applicability of default values across different datasets, the default values should be determined using datasets with diverse characteristics. Then, the method should be evaluated on datasets not used for determining these default values. Furthermore, in addition to reporting the performance with default values, the paper should include results after parameter-tuning all methods using cross-validation for each dataset. While the focus on simplicity is understandable, it does not justify limiting the scope of evaluation.

It appears that the paper suggests increasing tree depth when the number of input features is large. However, from the user’s perspective, this may not be desirable. For example, greedy algorithms can select important features during training, meaning that even with many features, the algorithm can automatically focus on a few important ones. The inability to leverage this advantage seems like a drawback. If the feature assignment algorithm in the proposed method addresses this concern, it should be explicitly demonstrated.

The experiments are conducted on very limited datasets. Although the datasets are drawn from various papers, the selection criteria are unclear. Given the availability of curated Tabular-Benchmark[1], it would be more convincing to use such benchmarks to avoid concerns that the selected datasets were chosen arbitrarily to produce favorable results.

Although comparisons are made between CART and GradTree, as noted in the Related Works section, there are many other methods that warrant comparison. Moreover, the core idea of the proposed method appears to be the feature ranking algorithm, which assigns features to nodes. It would be beneficial to evaluate this algorithm. For instance, comparisons with simple methods that assign features randomly to nodes or the application of a temperature-controlled softmax function to $\boldsymbol{w}_i$ in Equation (1), gradually lowering the temperature during training, could provide insights into the method’s effectiveness.

----

[1] Grinsztajn et al., Why do tree based models still outperform deep learning on typical tabular data? NeurIPS 2022 Datasets and Benchmarks Track

**Questions:**

- Is the proposed feature selection method superior to other methods? (See Weaknesses part for details)

- In practice, non-perfect binary tree structures such as decision lists or oblivious trees (where parameters are shared across nodes at the same depth) are often considered. How would the feature assignment algorithm in the proposed method adapt to such structures?

- The experiments do not appear to use standard benchmark datasets. How were the datasets in your paper chosen from the many available options? In particular, looking at the GitHub implementations, it seems that many more datasets are implemented than those presented in the paper. How were the 12 datasets mentioned in the paper selected from among the many available datasets?

- Figure 2 seems to show that increasing tree depth does not reduce training error. While this could be desirable from a generalization perspective, it suggests that the proposed method might not address the suboptimality associated with greedy algorithms. How do the authors interpret this result?

- The paper suggests increasing tree depth as the number of input features grows. However, from a user’s perspective, this may not be ideal. Even with a large number of features, a greedy algorithm can automatically select the few most important ones. How does the proposed method address this issue?

- While this paper focuses on building a single tree, it is also possible to extend the method to ensembles. Other methods, such as GradTree, have been extended to ensembles like GRANDE[2]. Would the proposed feature selection method work effectively in an ensemble setting as well?

----

[2] Marton et al., GRANDE: Gradient-Based Decision Tree Ensembles for Tabular Data, ICLR2024

---

### Official Review · Reviewer_rNYq · 2024-11-01

**Soundness:** 2
**Presentation:** 2
**Contribution:** 2
**Rating:** 3
**Confidence:** 3

**Summary:**

This work proposes a new probabilistic univariate decision tree approach. The core idea is to determine the splitting feature of each node in advance, and simplify the decision function at each node in order to achieve a reduced number of parameters. Additional experimental comparisons have been conducted on 12 benchmark datasets.

**Strengths:**

This paper proposes a simple axis-aligned soft tree approach, which may effectively help to avoid the issue of overfitting. As shown in the experimental results, the proposed approach shows better classification accuracy compared with previous methods. Besides, this paper also presents a detailed review of tree methods, and this could help readers better undersntand the background.

**Weaknesses:**

1) From my perspective, this work lacks sufficient innovation. The proposed method can be regarded as a special case of the typical soft decision tree, where the weight of decision function at each node is a pre-given one-hot vector. Specically, the proposed method determines the splitting feature of each node based on mutual information in advance. However, such approach may have following issues:
a)	As mentioned in the introduction, one important reason for introducing soft decision tree is to efficiently solve the optimal decision tree. However, the proposed method uses a highly heuristic approach to determine the splitting feature for each node, and such approach will undoubtedly converge to a suboptimal axis-algined soft tree. This contradicts the original motivation of the soft decision tree.
b)	The arrangement of features is also somewhat unreasonable. We take the decision tree in Figure 1 as an example. In the third layer, the relatively important feature x_3 is placed in the left subtree, while the less important features x_2 and x_1 are placed in the right subtree. It is hard to discern the intuition behind this.

2)	This work points out that the previous method, GradTree [Marton et al., 2024], requires a large amount of time and space.
a)	However, the article does not provide a comparison of the storage overhead between the proposed method and GradTree.
b)	Moreover, as shown in Table 6, the proposed method takes much more training time on several multi-class datasets to achieve comparable results. Besides, since reducing computational complexity is an important goal, Table 6 should be included in the main text rather than in the appendix.
From the perspective of experimental results, I think that the proposed method does not effectively address the shortcomings of previous methods.

3)	There are some minor issues of this paper：
a)	Lines 58-71 mention three methods, and the corresponding references should be provided.
b)	There is an extra bracket in the denominator of equation (1).
c)	The layout on page five is loose and should be written more concisely.

**Questions:**

1)	The experiments in Figure 2 and Figure 3 select CART as a baseline. Why not chooses the state-of-the-art method GradTree as a baseline?
2)	From Table 5, the F1 scores of GradTree appear to be lower than the results reported in its original paper on IRIS and SPLICE [Marton et al., 2024].

---

### Official Review · Reviewer_nDHf · 2024-11-01

**Soundness:** 2
**Presentation:** 3
**Contribution:** 3
**Rating:** 3
**Confidence:** 5

**Summary:**

This paper proposes ProuDT, a Probabilistic Univariate Decision Tree designed to enhance the accuracy, interpretability, and computational efficiency of traditional decision trees. Instead of fixed splits, the model assigns a single deterministic feature to each decision node, to ensure univariate splits while maintaining differentiability for gradient-based optimization. This approach reduces computational complexity by limiting the number of learnable parameters and enhances interpretability through transparent feature utilization. The authors conduct experiments on different datasets, showing that ProuDT outperforms other univariate and probabilistic decision trees.

**Strengths:**

- The study uses probabilistic splits that align with gradient-based optimization, contributing to improved classification accuracy without complex computation.
- The proposed method effectively addresses the limitations of traditional greedy decision trees and existing probabilistic trees by ensuring univariate splits and reducing computational complexity.
- By reducing the number of learnable parameters, ProuDT simplifies the optimization process, potentially improving scalability and ease of deployment.

**Weaknesses:**

- The authors rely on the preliminary study to justify key decisions, such as tree depth and feature ranking strategies. However, without presenting these results, it is not possible to verify the robustness of these choices or understand the specific conditions under which ProuDT performs optimally.
- The study uses a limited amount of datasets, which may not fully represent the diversity and complexity of real-world applications.
- Claims about enhanced interpretability are based on the univariate nature of the splits, as each node’s decision relies on one specific feature rather than a combination, which does suggest an enhancement on interpretability. However, the paper would benefit from quantitative metrics or experiments to substantiate these claims.
- There are instances, particularly in multi-class datasets, where performance falls short. Explicitly addressing these limitations would clarify ProuDT’s most suitable applications.

**Questions:**

- Can you elaborate on how sensitive ProuDT is to choices of hyperparameters like tree depth and the method of feature ranking, providing experimental results of your preliminary analysis?
- How do you measure the interpretability of ProuDT quantitatively? Including metrics could help substantiate claims regarding enhanced interpretability over existing methods.
- There is 39 datasets on your repository. Can you provide a rationale for presenting results for only 12 of them?

---

### Official Review · Reviewer_JfFU · 2024-11-02

**Soundness:** 2
**Presentation:** 3
**Contribution:** 2
**Rating:** 3
**Confidence:** 5

**Summary:**

The adaptation of soft-decision tree for univariate trees. "This is achieved by first setting up the tree structure and assigning features to each node based on their mutual information. Then, we optimize the thresholds (bias) and the values at the leaf nodes. The remaining part is the same as in the standard soft trees. Experiments on small scale datasets show some improvement over CART.

**Strengths:**

- it appears to be the first paper to suggest directly utilizing univariate splitting for probabilistic (soft) trees.
- simple and clear algorithmic framework.
- the resulting trees are interpretable due to specific choice of feature selection mechanism and balanced tree structure

**Weaknesses:**

- Limited novelty: I believe the main contribution here is the construction of the initial tree (cyclic assignment based on mutual information). There are potentially several other approaches which also should work decently (e.g. random feature assignment, warm-start from CART or other baseline). And the remaining part is standard training of soft trees.
- Experimental evidences are limited. Small to medium scale datasets are chosen for benchmarking where it is hard to judge who is a real winner. Indeed, performance gain over CART in most cases are statistically insignificant.
- Do we even need probabilistic (soft) tree formulation in this setup (and consequently SGD)? Note that the tree structure is fixed, feature allocation is fixed. Thus, I strongly believe that the MILP formulation will drastically simplify: we're gonna need binary variables only to keep track of instance assignment (which datapoint goes to which node). Consequently, MILP formulation here could be a strong (in terms of efficiency and performance) baseline to have. Especially for the given scale of datasets. In addition to this, some other non-greedy baseline is needed to fairly assess the performance. GradTree is clearly underforming w.r.t. CART and I wouldn't call it SOTA. Some other candidates seem to show stronger performance (TAO from Carreira-Perpinan & Tavallali, 2018).

**Questions:**

See weaknesses above and also:
- Datasets sizes and feature dimensions used in the experiments suggest that that shallow trees with balanced structure are sufficient in most cases. A lot of practical problems in industry require high imbalanced and quite complicated structure to get the best performance. Do you expect that this might be the problem with your proposed approach?
- Did you experiment with warm-starting your aglrotihm from CART or GradTree? I'd expect faster convergence but it would be interesting to see how it performs and whether it will get stuck in local optima...

---

### Comment · Area_Chair_8KsA · 2024-11-13
**authors - reviewers discussion open until November 26 at 11:59pm AoE**

Dear authors & reviewers,

The reviews for the paper should be now visible to both authors and reviewers. The discussion is open until November 26 at 11:59pm AoE.

Your AC

---

### Note · Authors · 2024-11-26

**Comment:**

We sincerely thank the reviewers for their time and thoughtful feedback on our submission. To improve our work, we will focus on clarifying the design and methodology to address potential misunderstandings and better communicate its contributions. Additionally, we plan to incorporate constructive suggestions to strengthen our results and enhance the presentation in future iterations.

We recognize that our approach, which prioritizes simplicity and ease of implementation while delivering strong performance, may not fully align with the expectations of some reviewers regarding innovation. Therefore, we have decided to withdraw our submission.

**Withdrawal Confirmation:**

I have read and agree with the venue's withdrawal policy on behalf of myself and my co-authors.